# A Dual-Branch Disentanglement Diffusion for ID-Attribute Conditional Face Generation

## Abstract

Face identity customization, i.e., face generation with specified identity, has received increasing attention owing to its extensive applications in personalized content creation. Although existing methods achieve high consistency in identity with reference faces, they still struggle to precisely manipulate fine-grained facial attributes. We attribute this issue to the inherent entanglement of identity and attribute information, as well as the lack of attribute-specific supervision. Accordingly, to address this issue, we propose AttPortrait, a high-quality identity-attribute conditional face generation framework. Based on a foundational face diffusion model, we introduce an extra disentanglement branch alongside the conventional denoising branch during the training stage. This extra branch employs explicit attribute supervision to encourage the model to capture the attribute information from the text prompts, effectively disentangling the identity and attributes and achieving precise attribute manipulation with high identity consistency. Comprehensive experiments demonstrate that our method achieves at least 34% improvement in attribute accuracy, attains identity similarity close to the state-of-the-art methods, and maintains comparable FID scores on real and synthetic datasets.

## 1 Introduction

Recent advances in diffusion models (Sohl-Dickstein et al., 2015; Ho et al., 2020; Song et al., 2021; Lipman et al., 2023; Albergo & Vanden-Eijnden, 2023; Liu et al., 2023), along with the availability of large-scale text-image pair datasets (Schuhmann et al., 2022; Changpinyo et al., 2021), have led to significant progress in text-to-image (T2I) generation (Rombach et al., 2022; Ramesh et al., 2022; Saharia et al., 2022; Balaji et al., 2022; Li et al., 2024a; Chen et al., 2024a; Esser et al., 2024; BlackForestLab; Xie et al., 2025). Correspondingly, face identity customization, as one of the key applications of T2I generation, has attracted growing research interest and reached new heights. Typically, existing methods incorporate facial features (Xiao et al., 2024; Li et al., 2024b; Wang et al., 2024b; Huang et al., 2024; Zhang et al., 2023a; Varanka et al., 2024) into diffusion models, enabling the specification of face identities when generating novel images.

Despite their success in specifying face identities, existing methods exhibit inherent limitations in specifying fine-grained facial attributes, such as hair style and age. As demonstrated in Figure 1, although the generated images have correct identities, their facial attributes do not match the given text prompts. In other words, the capacity to manipulate facial attributes via textual descriptions is substantially constrained when identity customization is required, which impedes the practical application of these approaches. We attribute this issue to the following two factors:

*1) The attribute information is entangled with the identity (ID) information.* Previous methods (Xiao et al., 2024; Valevski et al., 2023; Li et al., 2024b; Wang et al., 2024b; Huang et al., 2024) employ either general vision models such as CLIP image encoder (Radford et al., 2021) or specialized face recognition models such as ArcFace (Deng et al., 2019) to extract ID embeddings from the reference images for ID-conditional generation. However, these extractors usually fail to disentangle facial attribute information from the ID embeddings. In consequence, the attributes of the generated face images are often similar to those of the reference images, even when the text prompts specify different attributes. For example, as can be seen in Figure 1, both the reference and generated images display nearly identical facial attributes, demonstrating a high degree of entanglement between

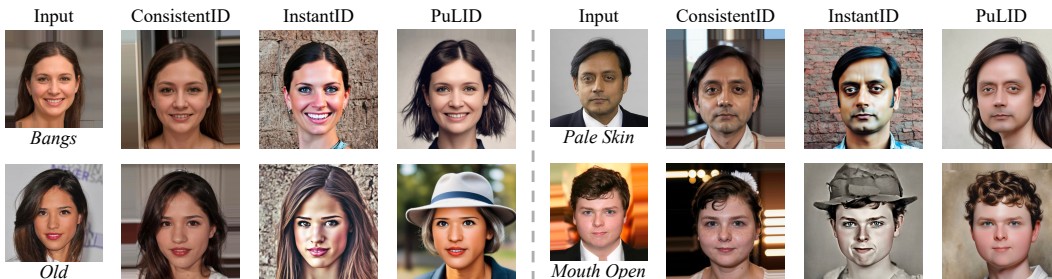

Figure 1: Given the reference images and the text description of the target attributes, the generated images of existing methods have the correct identity but fail to reflect the target attributes.

the attribute information and the ID information. *2) Previous methods lack explicit supervision to capture the attribute information from text prompts*, which is a more direct reason. Since there is no explicit supervision, most previous models tend to follow the entangled attribute in the ID embeddings rather than the attributes described in the text prompts when generating images.

In response to the aforementioned analysis, we propose **AttPortrait**, a face generation framework conditioned on both attributes and identity, which is capable of precise attribute control through textual descriptions while maintaining high identity consistency. Our framework consists of two branches: the denoising branch and the disentanglement branch, as shown in Figure 2. *The denoising branch* is a conditional diffusion model with classifier-free guidance (Ho & Salimans, 2022) that only employs an MSE loss, which is similar to existing methods (Cui et al., 2024; Xiao et al., 2024; Valevski et al., 2023; Chen et al., 2023). This branch is mainly responsible for generating high-quality and identity-consistent face images given the ID embeddings of reference faces. However, as discussed above, the attribute information is highly entangled with the ID information, which makes it difficult for the model to manipulate the attributes via text prompts. To solve this problem, we introduce an extra *disentanglement branch* with explicit supervision for attribute manipulation. Specifically, to ensure that the model faithfully captures the target attributes, we employ a pre-trained facial attribute predictor to assess the attributes present in the generated images. An attribute matching loss is then applied to minimize the discrepancy between the desired input attributes and those manifested in the generated images. Furthermore, we introduce a dual cross-attention module, which utilizes two parallel cross-attention blocks to separately incorporate the attribute information and the ID information. In this manner, the interference between the attribute information and the ID information is reduced, which improves both the attribute accuracy and identity consistency.

Our contributions are summarized below:

1. We reveal the entanglement of attribute and identity information in ID embeddings and quantify how this entanglement degrades attribute manipulation in diffusion models with comprehensive studies, providing critical guidance for future ID customization based on diffusion model.

2. We propose AttPortrait, an identity-attribute conditional generation framework, which adopts a denoising branch and a disentanglement branch to guide the generation process to ensure the correct attributes and identities. To our knowledge, AttPortrait is the first customization method that achieves satisfactory attribute manipulation with high identity consistency.

3. Extensive experiments demonstrate that our AttPortrait significantly outperforms existing approaches in attribute accuracy and attains identity consistency close to the state-of-the-art methods. Moreover, our model can achieve multiple attribute manipulation and zero-shot attribute manipulation with satisfactory performance.

## 2 RELATED WORK

### 2.1 SUBJECT-DRIVEN TEXT-TO-IMAGE DIFFUSION MODEL

Owing to the powerful generative capability of the text-to-image diffusion models (Rombach et al., 2022; Ramesh et al., 2022; Saharia et al., 2022; Balaji et al., 2022; Li et al., 2024a; Chen et al., 2024a; Esser et al., 2024; BlackForestLab; Xie et al., 2025), subject-driven methods have attracted increasing research attention. These methods aim to adapt large-scale diffusion models to synthesize images conditioned on specified subjects. Based on whether fine-tuning is required when testing a

new subject, these methods can be divided into two types: test-time fine-tuning methods (Ruiz et al., 2023; Gal et al., 2022; Dong et al., 2022; Kumari et al., 2023; Smith et al., 2023; Wang et al., 2024a; Yuan et al., 2023) and test-time free-tuning methods (Wei et al., 2023; Ye et al., 2023; Gal et al., 2023; Shi et al., 2024; Ma et al., 2024; He et al., 2024; Zhang et al., 2024). The test-time fine-tuning methods optimize the diffusion model at test time for each individual subject using one or more reference images. However, per-subject optimization is time-consuming and limits the applications. Test-time free-tuning methods, also known as encoder-based methods, typically incorporate diverse subject features into diffusion models with various fusion mechanisms. Specifically, in order to better capture subject-specific details, ELITE (Wei et al., 2023) inserts global subject features into the textual embedding and incorporates the local subject feature with an additional cross-attention, which inspires many subsequent works (Ye et al., 2023; Ma et al., 2024; Xiao et al., 2024; Shi et al., 2024). IP-adapter (Ye et al., 2023) is another representative subject-driven method, which introduces a lightweight adapter along with a separate cross-attention to fuse the subject information into the model and only finetune the lightweight adapter during training. To achieve better subject consistency, Subject-Diffusion (Ma et al., 2024) incorporates more subject information, including segmentations and bounding boxes, into the diffusion model and sets an adapter between self-attention and cross-attention.

## 2.2 ID Customization in Diffusion Model

A prominent direction within subject-driven methods is ID customization (Cui et al., 2024; Peng et al., 2024; Xiao et al., 2024; Huang et al., 2024; Li et al., 2024b; Wang et al., 2024b; Chen et al., 2024b; Wu et al., 2024b), which aims to generate images with specified identities. For example, FastComposer (Xiao et al., 2024) fuses multi-identity features with textual embeddings, and introduces localized attention control to support multi-identity generation. PortraitBooth (Peng et al., 2024) builds on FastComposer by improving the localized attention control through a truncated cross-attention mechanism, and additionally supports emotion control. Based on IP-Adapter (Ye et al., 2023), InstantID (Wang et al., 2024b) utilizes a modified ControlNet (Zhang et al., 2023b) to incorporate facial landmarks as an additional condition. Unlike the above methods that rely on a single reference image, PhotoMaker (Li et al., 2024b) combines multiple reference images of the same identity. IDAdapter (Cui et al., 2024) also employs multi-image design and further inserts adapters between attention blocks to better fuse identity features.

Moreover, to improve identity fidelity, several methods (Chen et al., 2023; Peng et al., 2024; Cui et al., 2024; Gal et al., 2024; Guo et al., 2024) introduce identity (ID) losses. PortraitBooth, IDAdapter, and PhotoVerse (Chen et al., 2023) directly generate images within one step at an early diffusion timestep and then calculate ID loss between the generated images and the reference images. However, such generated images are often noisy and low-quality, which reduces the effectiveness of the ID loss. To solve this issue, LCM-lookahead (Gal et al., 2024) and PuLID (Guo et al., 2024) employ fast sampling methods (Luo et al., 2023; Ren et al., 2024; Lin et al., 2024) to generate images of higher quality compared to previous approaches, which enables more reliable computation of ID loss. Although current approaches demonstrate strong identity consistency, the majority still fall short in correctly manipulating facial attributes.

## 3 Methods

### 3.1 Preliminaries

**Stable Diffusion** The Stable Diffusion model (Rombach et al., 2022) consists of three components: a CLIP (Radford et al., 2021) text encoder, a Variational Autoencoder (VAE) (Kingma et al., 2013), and a U-Net (Ronneberger et al., 2015). In the training phase, the VAE compresses the image $\mathbf{x}$ into the latent code $\mathbf{z}$. The latent code is subsequently perturbed by a Gaussian noise $\epsilon$. Then, the U-Net $\epsilon_\theta(\cdot)$ is optimized to denoise the noisy latent code $\mathbf{z}_t$, conditioned on the CLIP text embedding $\mathbf{e}$, with objective function as follows:

$$\mathcal{L}_{\text{diff}} = \mathbb{E}_{\mathbf{z}\sim\text{VAE}(\mathbf{x}),\epsilon\sim\mathcal{N}(\mathbf{0},\mathbf{I}),t,\mathbf{e}} \left[\|\epsilon - \epsilon_\theta(\mathbf{z}_t, t, \mathbf{e})\|^2\right]. \tag{1}$$

In the inference phase, a Gaussian noise $\mathbf{z}_T$ is iteratively denoised by the U-Net to obtain a clean latent $\mathbf{z}_0$, which is then decoded into the final image by the VAE decoder.

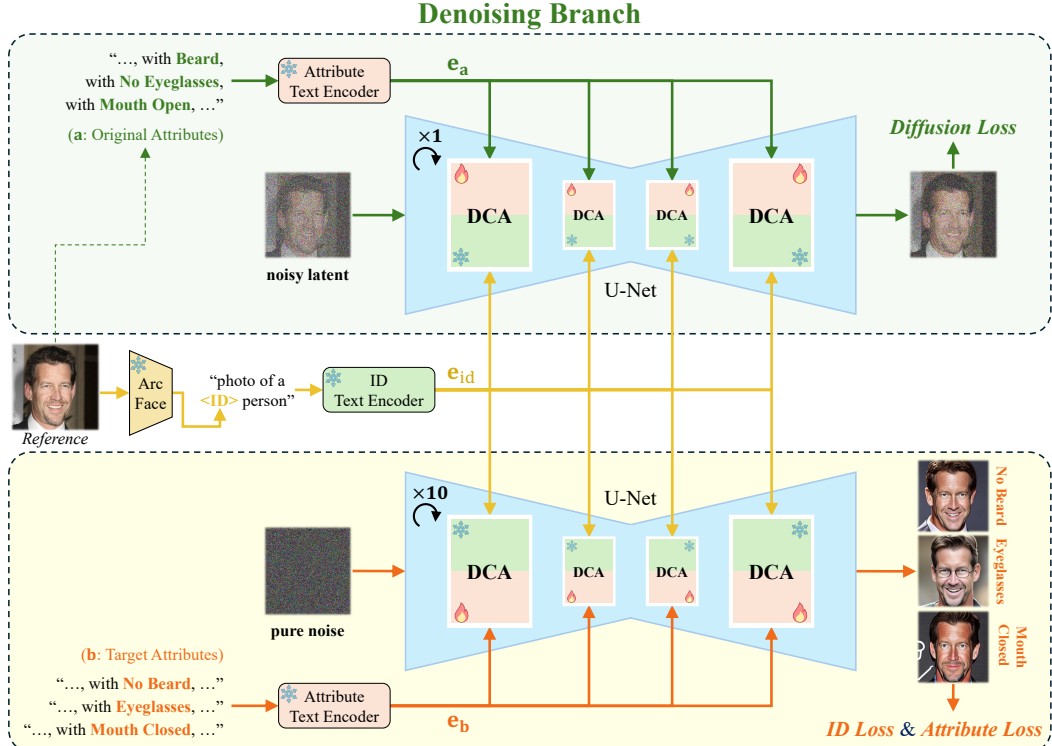

Figure 2: The overall framework of our AttPortrait. The upper half of this framework illustrates the denoising branch, which uses the original attribute embedding $\mathbf{e_a}$, containing attribute information present in the reference face. The lower part shows the disentanglement branch, which uses the target attribute embedding $\mathbf{e_b}$, containing target attribute information not present in the reference face. Both branches employ the identical ID embedding $\mathbf{e_{id}}$ and share the same attribute text encoder and U-Net. For clarity, textual descriptions are used to represent tokenized text embeddings.

**Embedding-Conditioning Cross-Attention Mechanism**   In the Stable Diffusion model, the U-Net uses multiple cross-attention (Vaswani et al., 2017) layers to incorporate the textual information, guiding the generation process according to the text prompt. Specifically, the text embedding $\mathbf{e}$ is projected to the key $K_\mathbf{e} = W_k\mathbf{e}$ and value $V_\mathbf{e} = W_v\mathbf{e}$, and the noisy latent code $\mathbf{z}_t$ is projected to the query $Q = W_q\mathbf{z}_t$. The cross-attention mechanism is formulated as follows:

$$Attn(Q, K_\mathbf{e}, V_\mathbf{e}) = \mathrm{Softmax}\left(\frac{QK_\mathbf{e}^T}{\sqrt{d}}\right)V_\mathbf{e}, \tag{2}$$

where $d$ is the dimension of $K_\mathbf{e}$, $V_\mathbf{e}$, and $Q$. In this work, we employ cross-attention to incorporate attribute information and identity information.

## 3.2   ID-ATTRIBUTE CONDITIONAL FACE GENERATION

**Task Definition**   Let $\mathbf{I_{ref}^a}$ denote a reference image with the target identity, and $\mathbf{a} = [a_1, \cdots, a_n]$ denote the corresponding attributes where each component $a_i$ is one kind of attribute, such as "Bangs", "Eyeglasses", and "Gender". Let $\mathbf{b} = [b_1, \cdots, b_n]$ denote the target attributes. Our objective is to develop a face diffusion model $G$ capable of generating faces with the target attributes and identities, which is formulated as follows:

$$\mathbf{I_{gen}^b} = G(\mathbf{b}, \mathbf{I_{ref}^a}), \tag{3}$$

where the generated image $\mathbf{I_{gen}^b}$ is expected to faithfully exhibit the target attributes $\mathbf{b}$ while maintaining the same identity as the reference image $\mathbf{I_{ref}^a}$.

To clarify, we use ArcFace (Deng et al., 2019) to extract the feature $\phi(\mathbf{I}_{\text{ref}}^{\mathbf{a}})$ from the reference image. This feature, following Arc2Face (Paraperas Papantoniou et al., 2024), is further mapped into an embedding $\mathbf{e}_{\text{id}}$, which is used as a condition input to the diffusion model. In the rest of the paper, to distinguish the two easily confused concepts, we refer to $\phi(\mathbf{I}_{\text{ref}}^{\mathbf{a}})$ as the "face recognition feature" and refer to $\mathbf{e}_{\text{id}}$ as the "ID embedding".

**Denoising Branch**  As shown in the upper branch of Figure 2, to generate identity-consistent and high-quality face images, we employ the classifier-free guidance (Ho & Salimans, 2022) diffusion model conditioned on the ID embeddings. Following Arc2Face (Paraperas Papantoniou et al., 2024), we first map the reference image $\mathbf{I}_{\text{ref}}^{\mathbf{a}}$ into the ID embedding $\mathbf{e}_{\text{id}}$. Besides, we employ CLIP to map the text description of the original attributes $\mathbf{a}$ into the attribute embedding $\mathbf{e}_{\mathbf{a}}$. Using the ID embedding and the original attribute embedding as a joint condition, our diffusion model learns to predict the noise using an MSE loss according to Eq. (1), which is formulated as follows:

$$\mathcal{L}_{\text{diff}} = \mathbb{E}\left[\|\epsilon - \epsilon_\theta(\mathbf{z}_t, t, \mathbf{e}_{\mathbf{a}}, \mathbf{e}_{\text{id}})\|^2\right]. \tag{4}$$

**Identity-Attribute Disentanglement Branch**  Although the denoising branch can generate identity-consistent faces, it fails to precisely manipulate the attributes due to the inherent entanglement between attribute and identity information within the ID embeddings. Specifically, the ID embedding $\mathbf{e}_{\text{id}}$ generally covers most of the attribute information, which makes the model directly ignore the attribute information from $\mathbf{e}_{\mathbf{a}}$. As a consequence, we cannot effectively control the attributes by modifying the attribute input.

To address this issue, as shown in the lower part of Figure 2, we introduce a disentanglement branch, which applies explicit supervision to encourage the model to capture the information from the attribute embeddings. Let $\mathbf{b}$ denote the target attributes, which is distinct from the original attributes $\mathbf{a}$ of the reference image $\mathbf{I}_{\text{ref}}^{\mathbf{a}}$. Given the target attributes and the reference identity image, in this branch, the diffusion model generates the final image $\mathbf{I}_{\text{gen}}^{\mathbf{b}} = G(\mathbf{b}, \mathbf{I}_{\text{ref}}^{\mathbf{a}})$ through 10 sampling steps, rather than just predicting the noise of one specific step. Further, to make the generated image accurately exhibit the target attributes, we apply an attribute matching loss as follows:

$$\mathcal{L}_{\text{att}} = \sum_{i=1}^{n} -b_i \log \mathcal{C}_i(\mathbf{I}_{\text{gen}}^{\mathbf{b}}) - (1 - b_i) \log(1 - \mathcal{C}_i(\mathbf{I}_{\text{gen}}^{\mathbf{b}})), \tag{5}$$

where $\mathcal{C}$ is a pretrained multi-attribute classifier and $\mathcal{C}_i$ is the prediction of the $i^{\text{th}}$ attribute. The objective function in Eq. (5) encourages the generated image $\mathbf{I}_{\text{gen}}^{\mathbf{b}}$ to be classified as possessing target attributes $\mathbf{b}$.

Besides, to avoid identity drift, we incorporate an auxiliary identity loss, formulated as follows:

$$\mathcal{L}_{\text{id}} = 1 - \frac{\phi(\mathbf{I}_{\text{gen}}^{\mathbf{b}}) \cdot \phi(\mathbf{I}_{\text{ref}}^{\mathbf{a}})}{\|\phi(\mathbf{I}_{\text{gen}}^{\mathbf{b}})\|\|\phi(\mathbf{I}_{\text{ref}}^{\mathbf{a}})\|}, \tag{6}$$

where $\phi$ is the face recognition feature from ArcFace (Deng et al., 2019). The objective function in Eq. (6) encourages a high identity similarity between the generated image and the reference image.

As mentioned above, we perform 10 sampling steps in this branch to generate the final image $\mathbf{I}_{\text{gen}}^{\mathbf{b}}$. However, backpropagation through 10 diffusion steps requires a large amount of GPU memory and a large number of FLOPs. To alleviate the computational requirements, we adopt DRTune (Wu et al., 2024a), which only maintains the gradient for a small subset of sampling steps while stopping the gradient of the U-Net input for the rest steps. In this manner, the attribute loss and identity loss can be effectively backpropagated through our model.

**Identity-Attribute Dual Cross-Attention**  To effectively capture both the attribute and identity information, we design a dual cross-attention (DCA) module. Specifically, as shown in Figure 2, the attribute embedding and the ID embedding are independently processed by distinct cross-attention blocks and then summed, formulated as follows:

$$Q_{\mathbf{a}}^* = Attn(Q, K_{\mathbf{e}_{\text{id}}}, V_{\mathbf{e}_{\text{id}}}) + Attn(Q, K_{\mathbf{e}_{\mathbf{a}}}, V_{\mathbf{e}_{\mathbf{a}}}), \tag{7}$$
$$Q_{\mathbf{b}}^* = Attn(Q, K_{\mathbf{e}_{\text{id}}}, V_{\mathbf{e}_{\text{id}}}) + Attn(Q, K_{\mathbf{e}_{\mathbf{b}}}, V_{\mathbf{e}_{\mathbf{b}}}), \tag{8}$$

where $Attn$ is defined in Eq. (2), $Q_{\mathbf{a}}^*$ and $Q_{\mathbf{b}}^*$ correspond to the DCA outputs in the denoising branch and disentanglement branch respectively. In this manner, the interference between the attribute and identity information is mitigated, thereby enhancing both the accuracy of attribute manipulation and the consistency of identity information.

**Objective Function**    The full objective function is given by:

$$\mathcal{L} = \mathcal{L}_{\text{diff}} + \lambda_{\text{att}}\mathcal{L}_{\text{att}} + \lambda_{\text{id}}\mathcal{L}_{\text{id}}, \tag{9}$$

where $\lambda_{\text{att}}$ and $\lambda_{\text{id}}$ are the hyperparameters to modulate the strength of the attribute loss and identity loss. In this paper, both $\lambda_{\text{att}}$ and $\lambda_{\text{id}}$ are set to 0.01.

## 4    EXPERIMENTS

### 4.1    SETUP

**Datasets**    Our training dataset consists of about 500K highest-quality faces selected from LAION-Face (Schuhmann et al., 2022) and their corresponding attribute labels predicted by our pretrained attribute predictor, including "Bald", "Young", "Male", "Bangs", "Black Hair", "Blond Hair", "Bushy Eyebrows", "Eyeglasses", "Mouth Slightly Open", "No Beard" and "Pale Skin". Each attribute is encoded as 1 if present and 0 if absent. For evaluation, we use two datasets with attribute labels, one is the 500 synthetic faces from Karras et al. (2020), referred to as Synth-test, the other is 5K faces randomly selected from CelebA test dataset (Liu et al., 2015), referred to as CelebA-test.

**Implement Details**    Our AttPortrait is built upon the SD-v1.5 architecture and uses two text encoders, namely ID text encoder and attribute text encoder, as illustrated in Figure 2. The ID text encoder and the U-Net are initialized from Arc2Face (Paraperas Papantoniou et al., 2024), and the attribute text encoder and the VAE are initialized from SD-v1.5. During the training phase, the denoising branch and the disentanglement branch utilize the shared U-Net and text encoders. Only the key and the value matrices in the cross-attention layers of attribute embeddings are trained with AdamW (Loshchilov & Hutter, 2017). In each iteration, the disentanglement branch generates images using a classifier-free guidance scale (Ho & Salimans, 2022) of 3.0 with 10 DPM-Solver (Lu et al., 2022) sampling steps, and for 8 randomly chosen steps we stop gradients of the U-Net input. During the inference phase, images are generated using the same classifier-free guidance scale with 25 DPM-Solver sampling steps. Our model is trained for 1 epoch on 8 NVIDIA A100 GPUs, with a constant learning rate of 1e-6 and a total batch size of 8.

### 4.2    EVALUATION PROTOCOLS

**Identity and FID Evaluation Protocol**    In this protocol, *Identity (ID) Similarity* and *FID* (Heusel et al., 2017) are employed to assess visual fidelity and identity consistency. Specifically, for each image, we randomly modify one attribute and generate a sample. This procedure is repeated twice. Then we report 1) the FID scores between the generated and reference sets, and 2) the average ID similarity for each generated–reference pair.

**Attribute Evaluation Protocol**    In this protocol, we use *Attribute Accuracy* to evaluate whether the generated images correctly reflect the target attributes. Specifically, for each attribute we modify it and generate two samples per reference image. Then we use the pretrained attribute predictor to predict the attribute of generated samples and compute the attribute accuracy, which is defined as the proportion of generated images where the target attributes are correctly present. Finally, we report the mean attribute accuracy over all attributes to show the overall performance.

### 4.3    COMPARISON WITH EXISTING METHODS

**Quantitative comparison**    We compare our AttPortrait with Arc2Face (Paraperas Papantoniou et al., 2024), ConsistentID (Huang et al., 2024), Photomaker (Li et al., 2024b), InstantID (Wang et al., 2024b), and PuLID (Guo et al., 2024) on our evaluation protocols. As shown in Table 1, AttPortrait achieves the highest attribute accuracy, with gains of **55.4**% on Synth-test and **34.4**% on CelebA-test. Our method also achieves the best identity similarity on Synth-test and the second-best on CelebA-test while maintaining comparable FID scores across both datasets. These results demonstrate that AttPortrait effectively manipulates attributes while maintaining identity consistency, surpassing prior approaches.

**Qualitative comparison**    Beyond the quantitative metrics, superior visual results of our AttPortrait are observed compared to existing approaches. As shown in Figure 3, other methods exhibit shortcomings to varying degrees. Arc2Face and ConsistentID generate high-quality faces but fail to control attributes. PhotoMaker is capable of controlling certain attributes, such as gender and

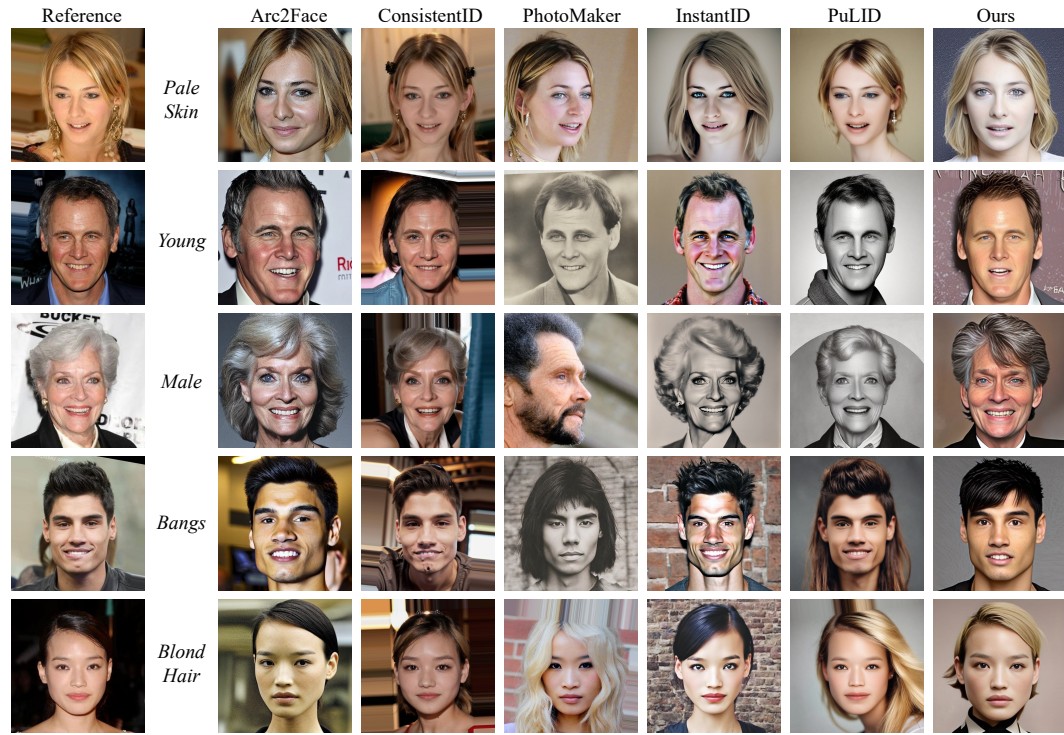

Figure 3: Qualitative comparison against existing works. We compare our method against Arc2Face, ConsistentID, PhotoMaker, InstantID, and PuLID across five distinct identities and five diverse attributes. For each row, we input the reference image and the text of the attribute on its right into these models. Here, we only compare the image quality against Arc2Face, since it does not support attribute text prompts.

hair color (3rd row and 5th row), but the outputs exhibit low identity consistency. InstantID usually generates less realistic faces (2nd row and 4th row) and shows limited ability to control attributes. PuLID can handle simple attributes like hair color, but it still has difficulty manipulating more abstract attributes such as age (2nd row) and gender (3rd row). In contrast, all images generated by AttPortrait exhibit high identity fidelity and accurately reflect the attributes specified in the texts, even under challenging cases like gender or age changes.

## 4.4 EXTENSIVE VISUAL RESULTS

**Multiple Attribute Manipulation** We show the performance of our AttPortrait in a more complicated scenario: multiple attribute manipulation, which imposes higher demands on attribute manipulation. The visual results in Figure 4 demonstrate that, despite manipulating more than one attribute, our approach maintains strong identity consistency.

Table 1: Quantitative comparison against existing methods on CelebA-test (Liu et al., 2015) and Synth-test (Karras et al., 2020). We report mean ID similarity (ID Sim), mean attribute accuracy (Att Acc), and FID. Here we do not report the attribute accuracy of Arc2Face (Paraperas Papantoniou et al., 2024) because it fails to accept the attribute text prompts.

| Method | CelebA-test | | | Synth-test | | |
| --- | --- | --- | --- | --- | --- | --- |
| | ID Sim ↑ | Att Acc ↑ | FID ↓ | ID Sim ↑ | Att Acc ↑ | FID ↓ |
| Arc2Face | **0.795** | - | 12.43 | 0.746 | - | 9.75 |
| ConsistentID | 0.499 | 0.094 | **7.99** | 0.475 | 0.074 | **3.98** |
| PhotoMaker | 0.234 | 0.525 | 9.01 | 0.249 | 0.357 | 5.21 |
| InstantID | 0.764 | 0.171 | 19.46 | 0.699 | 0.163 | 27.96 |
| PuLID | 0.631 | 0.371 | 9.56 | 0.609 | 0.322 | 5.16 |
| AttPortrait (ours) | 0.793 | **0.869** | 9.32 | **0.768** | **0.911** | 7.33 |

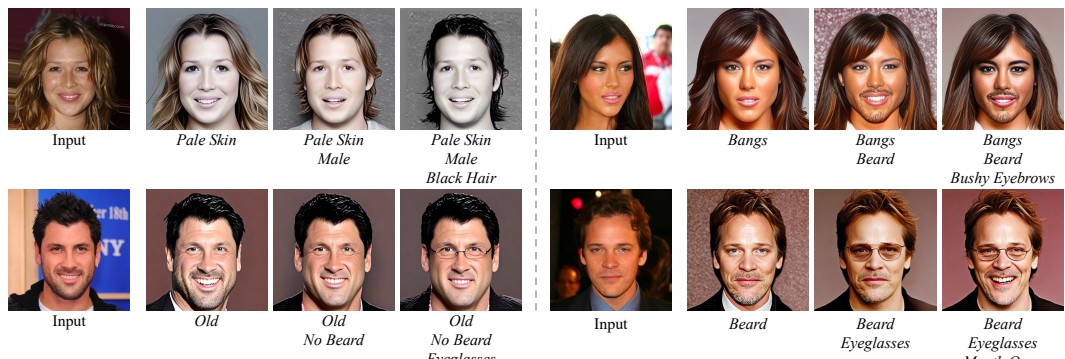

Figure 4: Visual results on multiple attribute manipulation.

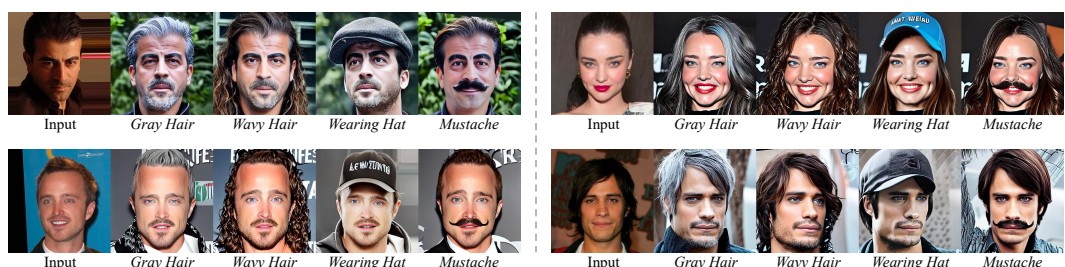

Figure 5: Visual results on zero-shot attribute manipulation.

**Zero-Shot Attribute Manipulation**    Our method supports zero-shot control of novel attributes. Even though certain attributes such as "Gray Hair", "Wavy Hair", "Wearing Hat", and "Mustache" are never seen during training, our model can still generate identity-consistent faces reflecting these attributes. Figure 5 illustrates several examples where novel attributes are correctly exhibited.

## 4.5    ABLATION STUDY

**Effect of Target Attributes**    In the disentanglement branch, target attributes distinct from the original ones are used as text prompts. To evaluate their impact, we compare the model trained with target attributes to that trained with original attributes. As shown in Figure 6 and Table 2, replacing target attributes with the original ones severely reduces the control capability over attributes, resulting in a marked decline in attribute accuracy.

**Effect of Disentanglement Branch**    As shown in Figure 6 and Table 2, removing the disentanglement branch substantially reduces attribute accuracy, increases FID scores, and even produces some distorted faces. These results highlight that the disentanglement branch is crucial for precise attribute manipulation while maintaining facial realism.

**Effect of Denoising Branch**    Figure 6 and Table 2 together show that removing the denoising branch results in severe noise and sharply higher FID scores on both datasets, highlighting its essential contribution to high-quality face generation.

**Effect of ID loss**    As shown in Figure 6 and Table 2, without the ID loss, the model suffers a marked drop in ID similarity and produces images with identity drift and noticeable distortions. These results clearly show that the ID loss effectively prevents identity drift during attribute manipulation.

**Effect of DCA**    To assess the impact of our dual cross-attention, we compare against a baseline where the ID and attribute embeddings are simply concatenated and injected through a single cross-attention block. As shown in Figure 6 and Table 2, removing the dual cross-attention reduces identity similarity, increases FID scores, and produces noisier images, highlighting the importance of our dual cross-attention mechanism in attribute manipulation while maintaining identity.

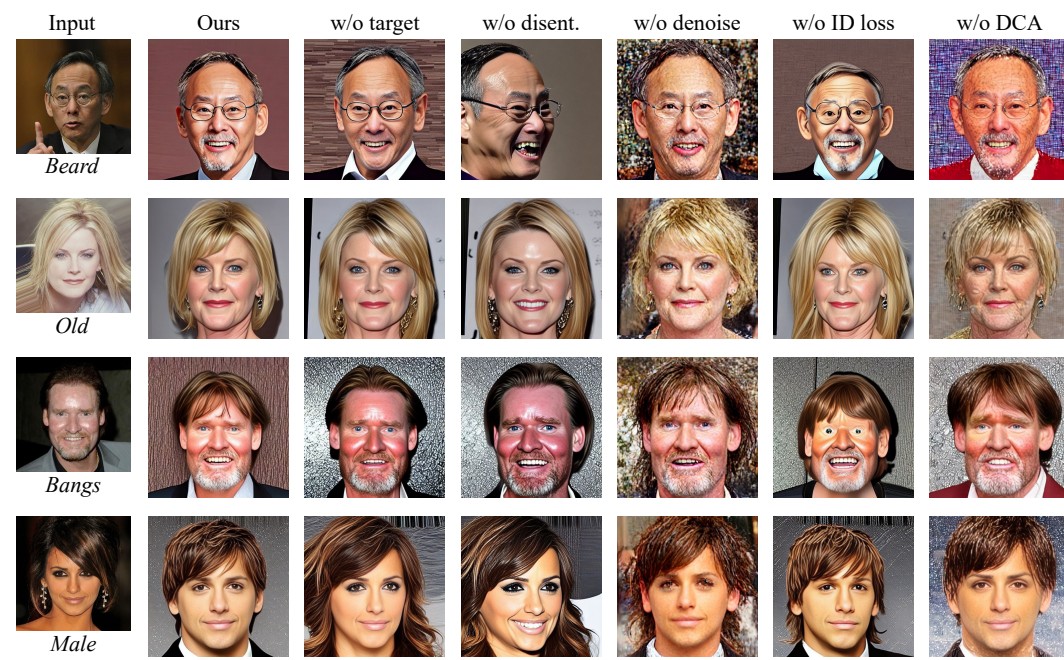

Figure 6: Qualitative comparison for ablation studies. For clarity, we denote "w/o target" as "w/o target attributes", "w/o denoise" as "w/o denoising branch", and "w/o disent." as "w/o disentanglement branch". Zoom in for better observation.

Table 2: Effect of each component, evaluated on CelebA-test[†] and Synth-test (Paraperas Papantoniou et al., 2024). Here, the CelebA-test[†] contains a randomly selected 1K images from CelebA-test (Liu et al., 2015).

| Ablations | CelebA-test[†] | | | Synth-test | | |
|---|---|---|---|---|---|---|
| | ID Sim ↑ | Att. Acc ↑ | FID ↓ | ID Sim ↑ | Att. Acc ↑ | FID ↓ |
| w/o Target Attributes | **0.810** | 0.481 | 13.31 | **0.788** | 0.520 | 9.75 |
| w/o Disentanglement Branch | 0.774 | 0.207 | 20.65 | 0.735 | 0.212 | 18.20 |
| w/o Denoising Branch | 0.805 | 0.932 | 24.99 | 0.772 | 0.949 | 26.17 |
| w/o ID Loss | 0.539 | **0.959** | 13.56 | 0.430 | **0.965** | 9.88 |
| w/o DCA | 0.694 | 0.886 | 28.35 | 0.660 | 0.902 | 28.74 |
| Full | 0.792 | 0.866 | **9.18** | 0.768 | 0.911 | **7.33** |

## 5 CONCLUSION AND LIMITATIONS

In this paper, we present AttPortrait, a face generation framework that precisely manipulates attributes through text prompts while maintaining high identity consistency. Existing methods fail to follow given attributes due to the entanglement of identity and attribute information in ID embeddings. However, we overcome this limitation by employing a novel dual-branch framework with explicit attribute supervision. Extensive experiments demonstrate that AttPortrait significantly outperforms prior methods in attribute accuracy. We hope our work can bring new inspiration to the area of personalized face generation.

While AttPortrait excels at identity consistency and precise attribute control, it has two main limitations. Firstly, our method exhibits a slight decline in image realism compared to Arc2Face (Paraperas Papantoniou et al., 2024). This may be caused by the attribute matching loss, which encourages the model to align with the given attribute, but also introduces a mild adversarial effect. Secondly, AttPortrait cannot generate full-body images with diverse backgrounds and styles, as it is trained exclusively on face-centric data. In the future, we plan to incorporate datasets with diverse backgrounds and contexts to enable simultaneous control over attributes, scenes, and styles.

## STATEMENTS

**Ethics Statement**  The ability of our model to generate realistic, attribute-controllable face images also carries risks such as disinformation, privacy violations, and misuse in identity fraud (Westerlund, 2019; Mirsky & Lee, 2021). To mitigate potential harm, we will release the model with mandatory watermarking, a clear usage license prohibiting malicious applications, and guidelines for explicitly labeling generated content. We encourage the community to follow ethical best practices and develop traceability mechanisms to discourage misuse.

**Reproducibility Statement**  Our method is fully reproducible. We provide a clear description of the proposed approach in section 3, and detailed experimental settings and implementation details are included in both section 4 and appendix A. In addition, we plan to release the source code to further facilitate reproduction of our results.

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

# APPENDIX

## A ADDITIONAL EXPERIMENT DETAILS

### A.1 DETAILS OF DATASET CONSTRUCTION

Our training dataset is selected from LAION-Face (Schuhmann et al., 2022) and contains approximately 500,000 face images. We start by using InsightFace (Deng et al., 2019) to detect and align faces in the original LAION-Face images. Next, we filter these faces and retain those with size larger than $133 \times 133$ pixels, face score above 0.85, and CLIP-IQA+ score (Wang et al., 2023) above 0.645, resulting in approximately 500,000 high-quality face images. We then employ GFPGAN v1.4 (Wang et al., 2021) to further enhance the image quality. Finally, we use a facial attribute predictor trained on CelebA (Liu et al., 2015) to assign attribute labels to each face.

### A.2 DETAILS OF THE INPUT PROMPTS

For existing methods, we follow the prompt templates described in the corresponding papers to guarantee their optimal performance. Specifically, given an attribute <att>, we use "photo of a person, <att>" for ConsistentID (Huang et al., 2024) and PhotoMaker (Li et al., 2024b), "<att>" for InstantID (Wang et al., 2024b), and "portrait, <att>" for PuLID (Guo et al., 2024). For our method, we construct the prompt with format: "who is " $+ \sum_{i=1}^{11}(\mathbf{M}(\mathrm{att}_i) + ", ")$, where $\mathrm{att}_i$ denotes the $i^{\mathrm{th}}$ attribute label and $\mathbf{M}$ is a label to text mapping shown in Table 3.

Table 3: Our mapping $\mathbf{M}$ from attribute labels to texts. N/A indicates an empty string.

| Attribute ($\mathrm{att}_i$) | Label = 1 | Label = 0 |
| --- | --- | --- |
| 1: Bald | "Bald" | N/A |
| 2: Young | "Young" | "Old" |
| 3: Male | "Male" | "Female" |
| 4: Bangs | "with Bangs" | N/A |
| 5: Black Hair | "with Black Hair" | N/A |
| 6: Blond Hair | "with Blond Hair" | N/A |
| 7: Bushy Eyebrows | "with Bushy Eyebrows" | N/A |
| 8: Eyeglasses | "with Eyeglasses" | N/A |
| 9: Mouth Slightly Open | "with Mouth Slightly Open" | N/A |
| 10: No Beard | "with No Beard" | N/A |
| 11: Pale Skin | "with Pale Skin" | N/A |

## B ADDITIONAL VISUAL RESULTS

We present additional qualitative comparisons of single attribute manipulation in Figure 7 and Figure 8, which demonstrate that our AttPortrait achieves more effective attribute control than existing methods while maintaining high identity consistency. We also present additional results of multi-attribute manipulation in Figure 9, which accurately reflect the given attributes, demonstrating our satisfactory ability to handle multiple attributes.

## C FAILURE CASES

Three types of failure cases of our model are shown in Figure 10. The first is visual deformation, where the generated faces exhibit irregularities or distorted features that reduce visual realism. The second is unexpected artifacts, such as colored marks or undesired accessories. These two types of failure might be related to adversarial effects (Szegedy et al., 2013; He et al., 2019) caused by the attribute matching loss. The third is ineffective manipulation, where the given attributes are not correctly reflected in the generated faces.

## D THE USE OF LARGE LANGUAGE MODELS

The use of LLMs in this paper was strictly limited to enhancing readability, such as refining language and correcting grammatical errors. No part of the research process, including problem formulation, method design, experimental implementation, or result interpretation, was assisted by LLMs.

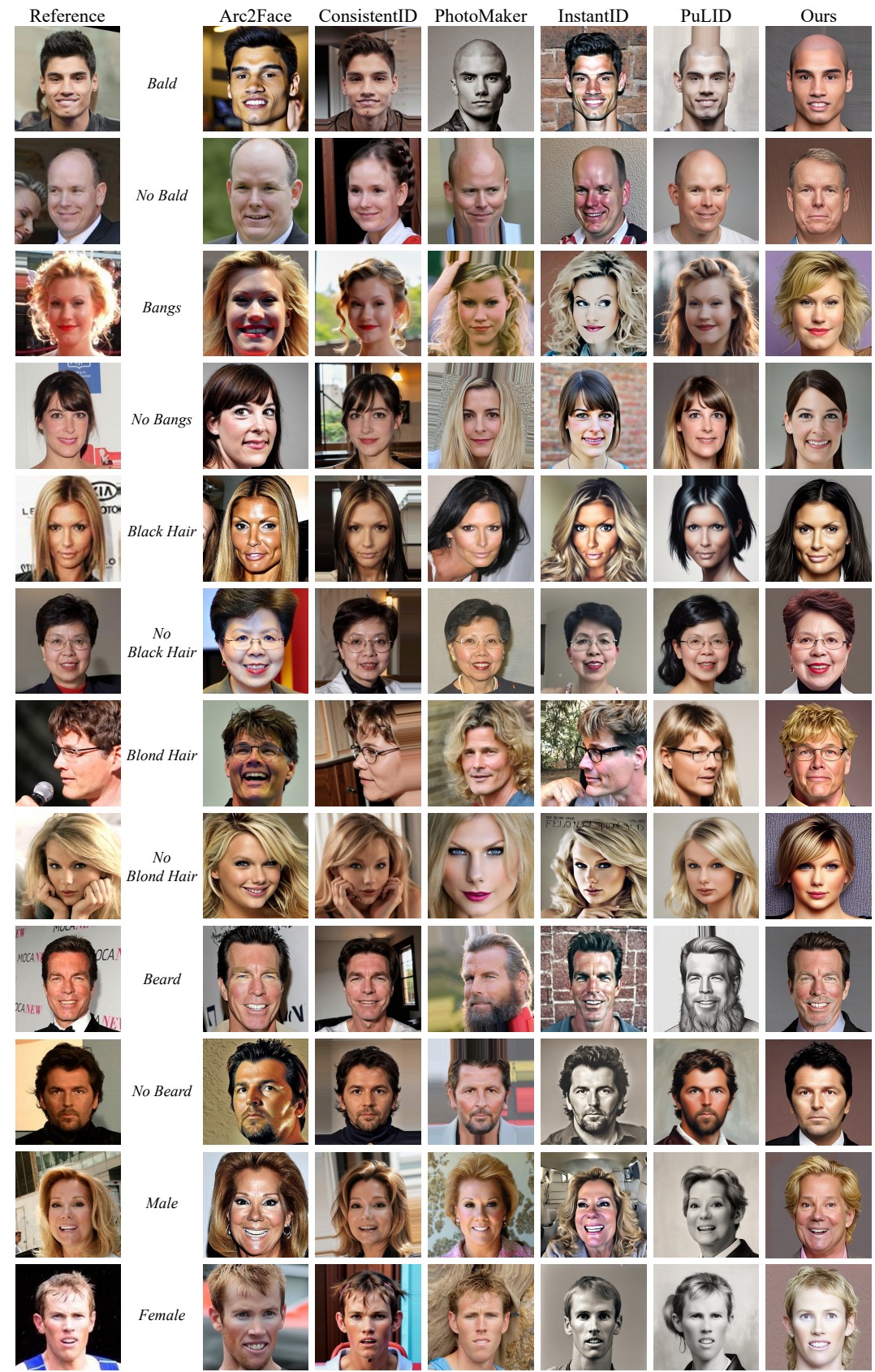

Figure 7: Additional qualitative comparisons (1/2). The results of Arc2Face (Paraperas Papantoniou et al., 2024) are included solely to show the image quality and identity consistency, as it does not support attribute manipulation.

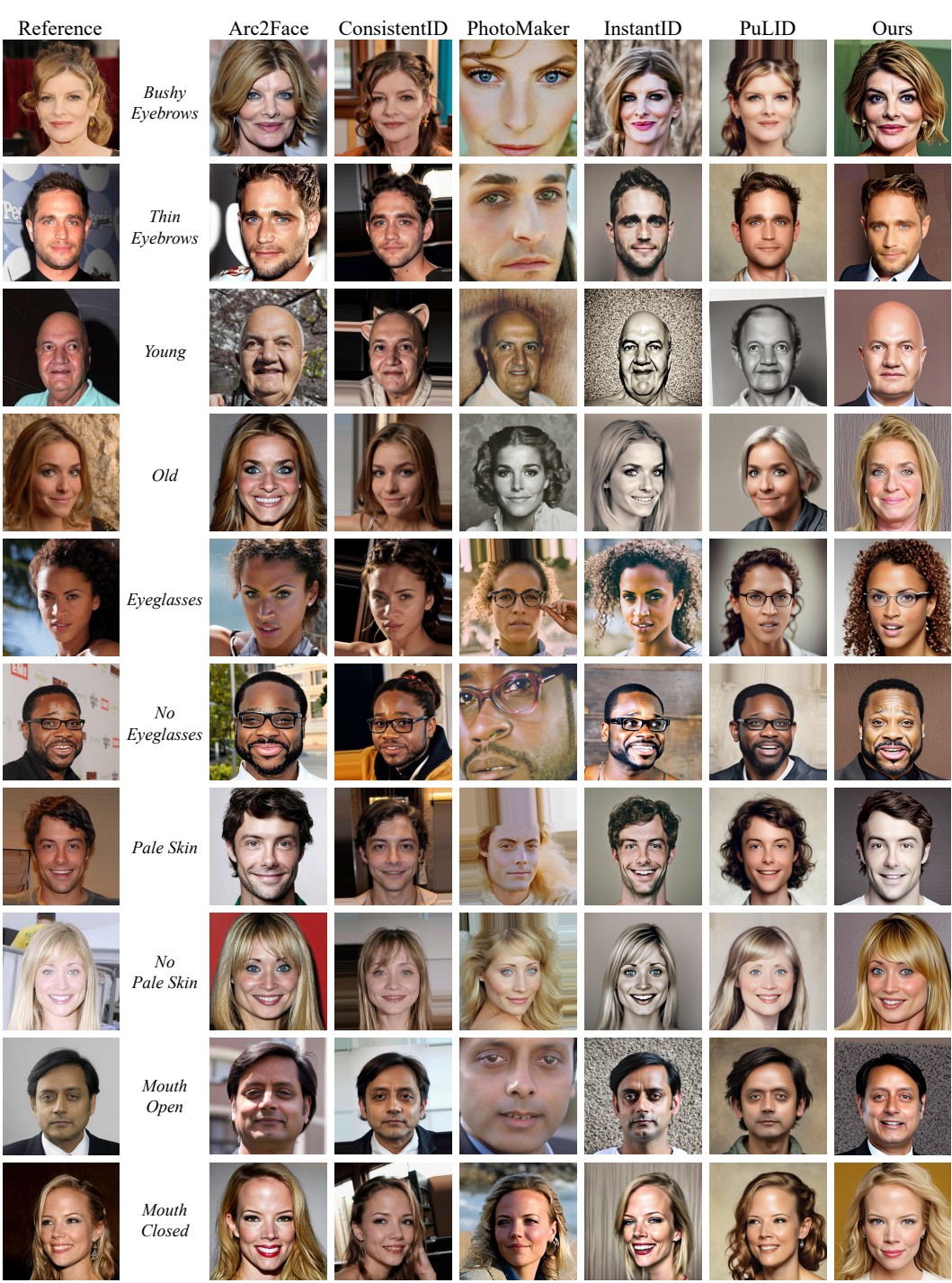

Figure 8: Additional qualitative comparisons (2/2). The results of Arc2Face (Paraperas Papantoniou et al., 2024) are included solely to show the image quality and identity consistency, as it does not support attribute manipulation.

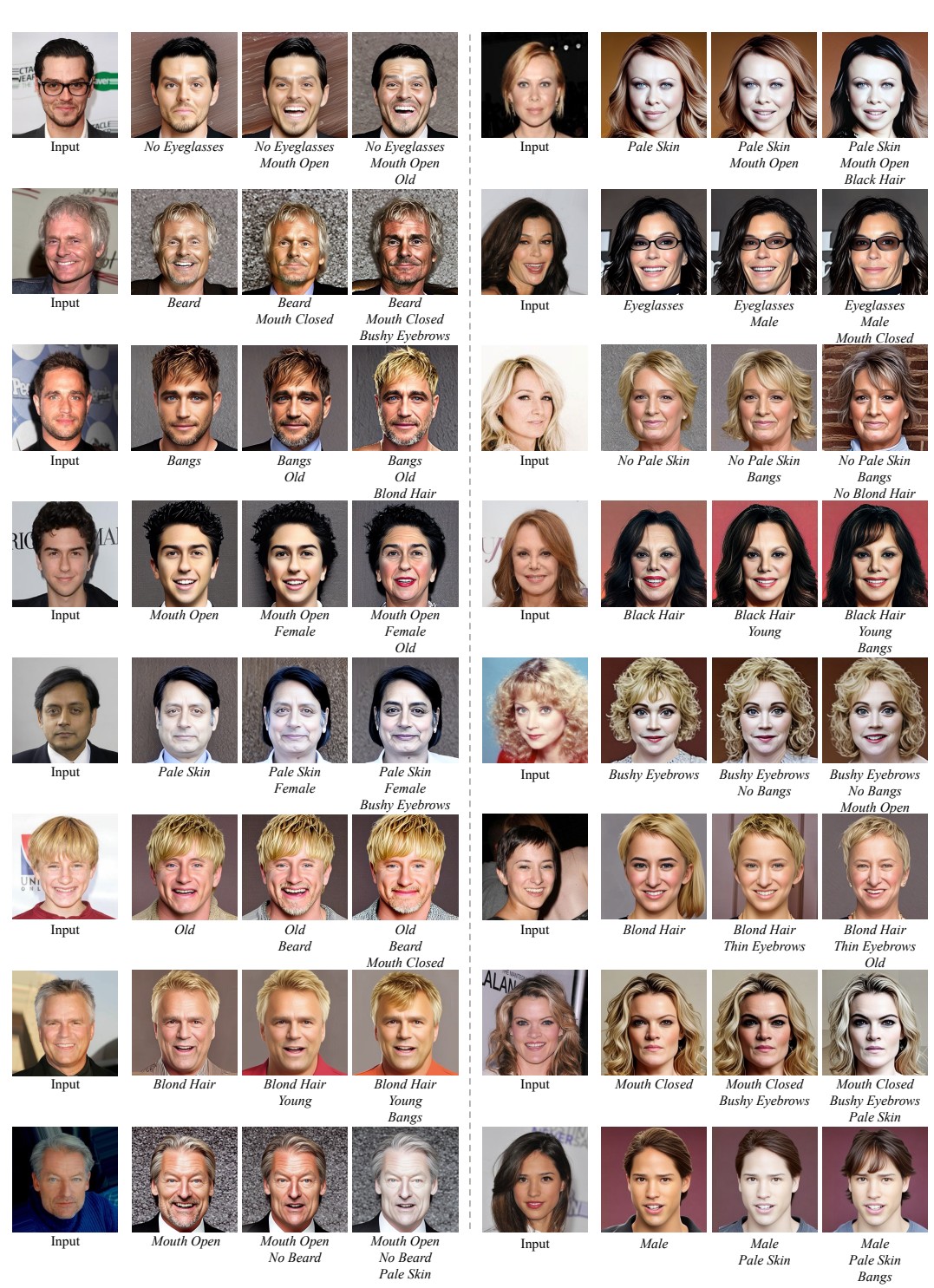

Figure 9: Additional results of multi-attribute manipulation.

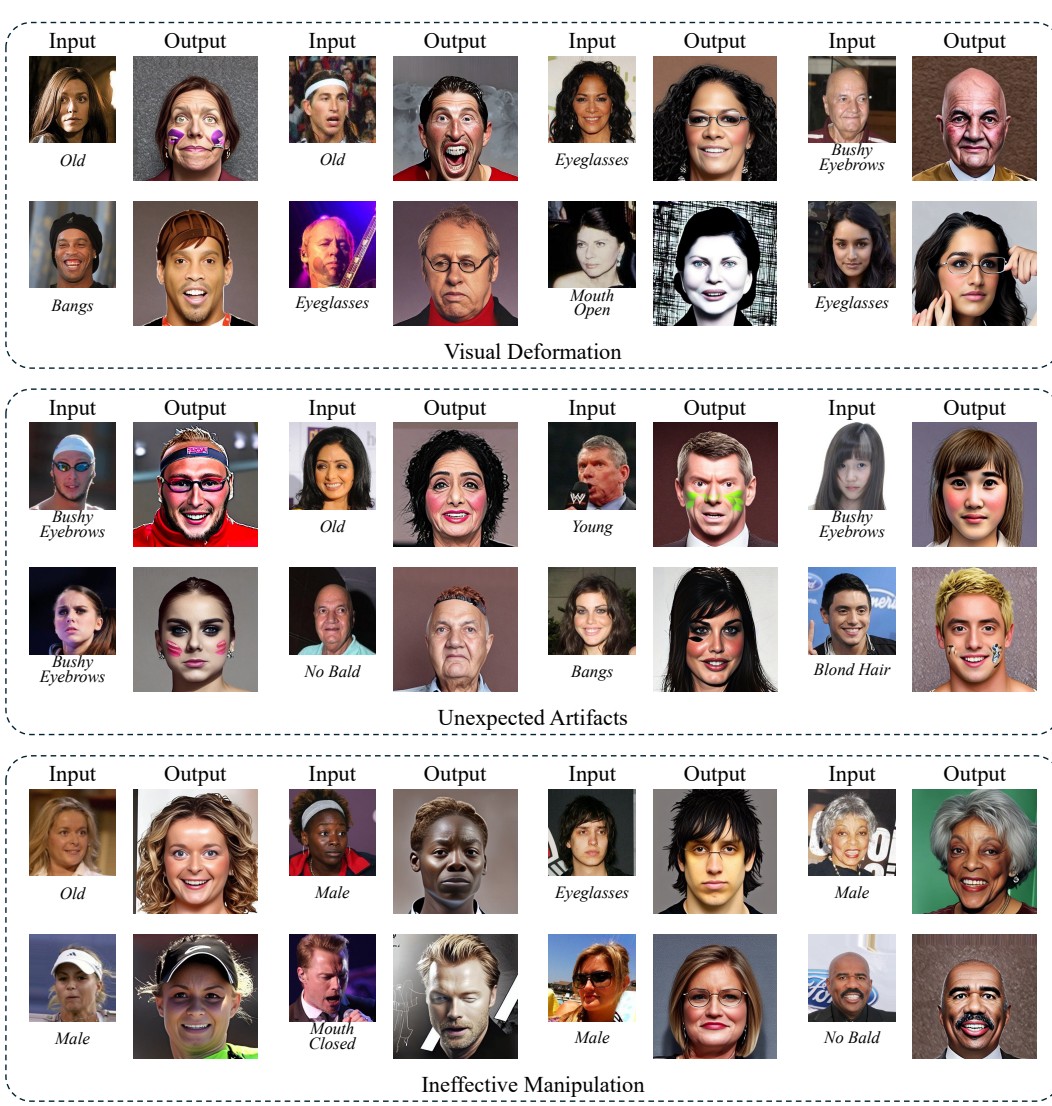

Figure 10: Failure cases.

