# OpenReview forum: "A Dual-Branch Disentanglement Diffusion for ID-Attribute Conditional Face Generation"
_ICLR.cc/2026/Conference — ICLR 2026 Conference Withdrawn Submission_

### Official Review · Reviewer_oERR · 2025-10-16

**Soundness:** 2
**Presentation:** 3
**Contribution:** 2
**Rating:** 6
**Confidence:** 3

**Summary:**

This paper proposes AttPortrait, a dual-branch diffusion framework for identity-attribute conditional face generation. In addition to the standard denoising branch, a disentanglement branch with explicit attribute supervision is introduced to decouple identity and attribute representations.
The method improves attribute accuracy over previous approaches, but still have challenges for improvement in realism and the comparison with baseline.

**Strengths:**

・Clearly formulates and analyzes the identity–attribute entanglement problem in diffusion-based face generation.

・The dual-branch design effectively separates identity and attribute information, improving fine-grained attribute control.

・The paper is well-structured and easy to follow, with clear motivation.

**Weaknesses:**

・The attribute set used for training and testing only covers a few simple binary traits like bangs, beard, or eyeglasses. That makes it hard to really judge how well the model handles identity drift or disentanglement in general. It would help to include more diverse attributes like expression, pose, or lighting for a fuller evaluation.

・While the model gets better scores on attribute accuracy, some generated faces look a bit too smooth or have that “oil-painting” texture. It feels like there’s a trade-off between keeping attributes accurate and making the image look natural.

・Most of the experiments are done on Arc2Face-based datasets, so it’s not clear how well the method generalizes. Testing on other datasets like DCFace or CemiFace would make the results more convincing.

・Some tricky attributes, especially ones tied closely to identity like Beard or Old, still seem to carry over from the reference image instead of following the target condition. That suggests the disentanglement isn’t fully working yet.

・The model depends on a pretrained attribute classifier for supervision, so any bias or label errors there might affect training. The paper doesn’t really look into how that could impact fairness or robustness, which might be worth mentioning.

**Questions:**

See Weaknesses

---

### Official Review · Reviewer_WYGJ · 2025-10-30

**Soundness:** 2
**Presentation:** 3
**Contribution:** 2
**Rating:** 4
**Confidence:** 4

**Summary:**

This work aim to improve facial attribute
manipulation through text prompt when generating customized ID-preserving face images in face image generation models. Two-branch training scheme is proposed for enhancing attribute manipulation. Attribute information and ID information are separately encoded in cross-attention layers for training.

**Strengths:**

1. This paper is well-written and easy to follow.
2. Authors proposed a dual cross attention (DCA) to learn ID and attribute embedding separately.

**Weaknesses:**

1. Author used own pretrained attribute predictor to generate label for training images. However, there are only limited types of attributes.

2. In Identity and FID Evaluation Protocol, author only generated image with single attribute manipulation for measuring ID similarity and FID. It cannot thoroughly measure ID-preserving ability of model. Moreover, it is also necessary to measure SSIM.

**Questions:**

1. Existing face generation models e.g., Arc2Face, InstantID, they are trained with fixed prompt.
Can attribute manipulation being enhanced by simply fine-tuning the model with prompts that describing various facial attributes?
2. How efficient of proposed method in terms of training time, GPU memory and inference speed?
3. For existing methods, author omitted the work IPA-FaceID. Why this method is not included in comparison?
4. In Identity-Attribute Disentanglement Branch,
author define attribute matching function loss function using pre-trained multi-attribute classifier. Is this the same one that used in evaluation experiment?

---

### Official Review · Reviewer_Gu6z · 2025-11-01

**Soundness:** 2
**Presentation:** 3
**Contribution:** 2
**Rating:** 2
**Confidence:** 5

**Summary:**

This paper introduces a diffusion-based framework for identity-attribute conditional face generation. Authors aim to address the entanglement of identity and attribute information in ID embeddings. During training, a pre-trained attribute classifier provides supervision for attribute accuracy, and an ID similarity loss maintains identity consistency.

**Strengths:**

The motivation of attribute–identity entanglement is compelling and authors its impact through both visualization and ablation.

Experiments on CelebA-test and Synth-test show that AttPortrait achieves a large improvement in attribute accuracy.

**Weaknesses:**

The idea to disentangle of identity and attribute information has been a long-time research, thus the contribution of this work seems incremental.

The authors compared with many approaches from before 2020, which is insufficient. It is recommended to include comparisons with more recent generation works.

In the ablation study, the full component does not show a clear advantage compared to w/o Target Attributes and w/o ID Loss, thus the final design seems more like a trade-off.

**Questions:**

please refer to weakness

---

### Official Review · Reviewer_j6mv · 2025-11-02

**Soundness:** 2
**Presentation:** 2
**Contribution:** 1
**Rating:** 2
**Confidence:** 4

**Summary:**

This paper proposes AttPortrait, a diffusion-based framework for face generation that aims to disentangle identity and attribute information. The method introduces two branches during training: 1) a denoising branch to ensure high-quality and identity-consistent synthesis, and 2)
a disentanglement branch that uses an explicit attribute matching loss guided by a pretrained attribute classifier.
A dual cross-attention (DCA) mechanism is also presented to separate ID and attribute signals in the U-Net. The model reportedly improves attribute accuracy by over 30% compared to existing ID-conditional diffusion baselines while maintaining identity similarity and FID performance.

**Strengths:**

1) Clear motivation: The paper identifies a valid shortcoming in identity-conditioned diffusion models — limited control over fine-grained attributes. 2) Readable presentation: The method, architecture diagrams, and equations are well-organized and clearly explained.
3) Comprehensive comparisons: The authors evaluate against several strong baselines (Arc2Face, InstantID, PuLID, etc.) and perform ablation studies.

**Weaknesses:**

Limited novelty / Incremental contribution
The proposed framework is largely an engineering combination of existing techniques rather than a fundamentally new approach.
The “dual-branch” structure mirrors common multi-loss or auxiliary supervision setups.
The “dual cross-attention” is simply two separate cross-attention blocks summed together; this is a minor architectural tweak.
The attribute matching loss is a straightforward supervised signal using an off-the-shelf classifier.
Overall, the contribution feels incremental relative to Arc2Face, IDAdapter, ID3 and PhotoMaker.

A similar concept of ID/attribute was already presented in ID3.

Attribute accuracy is computed using the same classifier that provides supervision — this invalidates claims of improvement and introduces strong confirmation bias.


Jianqing Xu, Shen Li, Jiaying Wu, Miao Xiong, Ailin Deng, Jiazhen Ji, Yuge Huang, Guodong Mu, Wenjie Feng, Shouhong Ding, Bryan Hooi:
ID3: Identity-Preserving-yet-Diversified Diffusion Models for Synthetic Face Recognition. NeurIPS 2024

**Questions:**

Since the attribute predictor used for supervision is also used for evaluation, how can we be confident that the improvements reflect true controllability rather than overfitting to the same classifier’s feature space?
Have the authors tried evaluating with an independent attribute classifier?

The DCA appears to be two standard cross-attention modules in parallel. Could the authors clarify what distinguishes this from existing multi-condition attention or modulation-based fusion used in prior works like PhotoVerse or IP-Adapter?

How does the model handle contradictory attributes (e.g., “young” + “gray hair”  or "Woman" + "mostash"?

---

### Note · Authors · 2025-11-18

I have read and agree with the venue's withdrawal policy on behalf of myself and my co-authors.